# Functional and Compositional Changes in Ileal Microbiota in Piglets During the Nursing Period Revealed by 16s rRNA Gene and Metagenomics

**DOI:** 10.3390/ani15213102

**Published:** 2025-10-25

**Authors:** Boxuan Yang, Pengfei Shen, Zhijian Xu, Jianbo Yang, Bo Song, Hui Jiang, Jianmin Chai, Jiangchao Zhao, Feilong Deng, Ying Li

**Affiliations:** 1Guangdong Provincial Key Laboratory of Animal Molecular Design and Precise Breeding, School of Animal Science and Technology, Foshan University, Foshan 528225, China; boxuanyang04@163.com (B.Y.); shenpengfei0909@163.com (P.S.); xzj13286551088@163.com (Z.X.); jianbo952@gmail.com (J.Y.); songbo@fosu.edu.cn (B.S.); jianghui1001@fosu.edu.cn (H.J.); jchai@uark.edu (J.C.); 2College of Animal Science, South China Agricultural University, Guangzhou 510642, China; jzhao77@uark.edu

**Keywords:** metagenome-assembled genomes, pig gut microorganism, nursing piglets, 16s rRNA gene sequence, ileum

## Abstract

**Simple Summary:**

This study explored how the gut microbes in the ileum (a part of the small intestine) of piglets develop and function during the nursing period. Using 16s rRNA gene sequencing methods, we found that the diversity of microbial species in the ileum remained relatively stable. *Lactobacillus* became dominant from day 7 onward, with specific species like *Lactobacillus johnsonii* and *Limosilactobacillus pontis* appearing at different stages. Functional analysis revealed that by day 21, using metagenomics analyses, the ileal microbes were especially active in butanoate metabolism. By day 28, pathways related to cysteine, methionine, and lysine metabolism became more prominent—processes vital for amino acid synthesis—primarily associated with *Limosilactobacillus mucosae*, *Limosilactobacillus oris*, and *Limosilactobacillus pontis*. These findings show that microbial community and functions change significantly between day 21 and 28, indicating a key developmental phase for gut health in piglets.

**Abstract:**

In piglets, the gut microbiota matures in a segment-specific manner during the nursing period, while fecal-based studies provide limited functional resolution across intestinal sites. We profiled the ileum using 16s rRNA gene sequencing and assessed segmental functions by shotgun metagenomics at selected ages. Ileal species richness and diversity were relatively stable across days. *Lactobacillus* were prominent from day 7, with stage-associated taxa including *Lactobacillus johnsonii*, *Lactobacillus delbrueckii*, *Ligilactobacillus salivarius*, and *Limosilactobacillus pontis*. Through metagenomic functional analysis, at 21 days, genes were enriched in butanoate metabolism, and *Limosilactobacillus pontis* as a potential probiotic played an important role in it. At day 28, metagenomic analysis indicated higher relative abundance in the ileum of pathways linked to cysteine and methionine metabolism and lysine biosynthesis, largely carried by *Limosilactobacillus mucosae*, *Limosilactobacillus oris*, and *Limosilactobacillus pontis*. These data describe the composition and function of the ileum in the intestines of piglets and indicate a differentiation period around day 21 to day 28.

## 1. Introduction

The gut microbiota is fundamental to immune maturation and overall health in piglets [1,2]. Colonization initiates immediately after birth and proceeds through a phase of rapid ecological succession from 7 to 21 days, with increased compositional stability approaching weaning at approximately 21 to 30 days [3,4,5,6]. Beyond temporal maturation, pronounced spatial heterogeneity characterizes the intestine. The small intestine exhibits fast luminal transit and relatively higher oxygen tension, whereas the large intestine provides an anaerobic, fermentative niche that yields short-chain fatty acids, including acetate, propionate, and butyrate [7]. Within this division of labor, the ileum, located between the jejunum and cecum, contributes to the breakdown of complex carbohydrates, energy harvest, and regulation of mucosal immunity [8,9,10].

Most prior investigations have relied on fecal sampling and single time points, which limit the resolution of segment-specific and time-dependent processes during early life [11,12,13,14]. During nursing, piglets experience physiological, environmental, and social stressors that can modulate microbial assembly [15,16,17], yet interactions between these pressures and segmental niches remain insufficiently defined. Segment-resolved and time-resolved profiling is therefore required to delineate early developmental trajectories.

This study sampled piglets at five developmental ages (day 1, 7, 14, 21, and 28) and profiled ileal communities using 16s rRNA gene sequencing to chart dynamics. A two-tier design incorporated shotgun metagenomics at day 14–21 and 21–28 to compare gene repertoires and KEGG-annotated pathways. The approach accommodated low microbial biomass in early life and the higher DNA input typically required for shotgun metagenomics relative to 16s rRNA gene profiling [18,19,20,21]. The working hypothesis posited a differentiation window from day 14 to 21 and 21 to 28 during which ileum functional capacities diverge with the enrichment of amino acid metabolism in the ileum and antioxidant-related processes. This framework aims to provide segment-resolved, time-resolved references for early-life microbiota in piglets and to inform subsequent mechanistic and intervention studies.

## 2. Methods

### 2.1. Sample Collection

This study used a crossbred piglet model (DLY Duroc × Landrace × Yorkshire) under uniform feeding conditions in Guangzhou, China (the entire breeding period was managed in accordance with the regular procedures of the pig farm). Under the guidance of professional veterinarians, piglets were slaughtered at five time points, 1, 7, 14, 21, and 28 days of age, and intestinal content samples were collected and then used for analyzing the temporal dynamics of the ileum, with six piglets per time point (a total of thirty piglets). To ensure consistency, piglets with similar body weights from different stalls were selected for slaughter, maintaining a 1:1 male-to-female ratio (to reduce the need for additional repetitive experiments due to extreme weight or gender differences and to minimize stress and pain to the greatest extent possible). Upon dissection, the ileum was carefully separated. After separating the middle section of the ileum, we gently squeezed the intestinal content into centrifuge tubes (Servier (Beijing) Pharmaceutical R&D Co., Ltd., Beijing, China) to avoid contact with oxygen. Immediately following collection, the intestinal contents were placed on liquid nitrogen for rapid freezing.

### 2.2. Library Construction, Quality Control, and Sequencing

Sequencing libraries were generated, and indexes were added to attribute sequences to each sample. Briefly, PCR amplification of targeted regions was performed by using specific primers connecting with barcodes. The PCR products with proper size were selected by 2% agarose gel electrophoresis. The same amount of PCR products from each sample was pooled, end-repaired, A-tailed, and further ligated with Illumina adapters. Libraries were sequenced on a paired-end Illumina platform to generate paired-end raw reads. Subsequently, the library quality was assessed and quantified by qPCR. Quantified libraries were pooled and sequenced, according to the effective library concentration and data amount required, by Novogene Bioinformatics Technology Co., Ltd. (Beijing, China).

### 2.3. The 16s rRNA Gene Sequence Analysis

The raw sequencing data were processed using QIIME2 (version 2023.9.1) [22]. First, the data were imported into QIIME2 with the import format set to PairedEndFastqManifestPhred33. After import, the V3-V4 region primers were trimmed (forward: ACTCCTACGGGAGGCAGCA; reverse: GGACTACHVGGGTWTCTAAT), followed by the merging of the paired-end reads. Quality filtering was performed by removing reads with a quality score lower than 30 in any 5 bp sliding window, resulting in high-quality reads. The high-quality reads were then processed using the QIIME2 Deblur plugin for quality control, including filtering, dereplication, chimera removal, and the generation of a feature table and representative feature sequences. For taxonomic assignment, representative sequences were classified with the Greengenes 13_8 99% reference database [23] (V3–V4 region). Statistical tests for alpha (Simpson) and beta (Bray–Curtis) diversity were conducted using RStudio (version 4.4.2, 2024). The identification of characteristic microorganisms was achieved through LEfSe (Linear Discriminant Analysis Effect Size) analysis [24].

### 2.4. Metagenomic Library Preparation

A total amount of 0.2 μg DNA per sample was used as input material for the DNA library preparations. Briefly, genomic DNA sample was fragmented by Covaris LE220R-plus (Covaris, Woburn, MA, USA) to a size of 350 bp. Then DNA fragments were end-polished, A-tailed, and ligated with the full-length adapter for Illumina sequencing, followed by further PCR amplification. PCR products were purified by AMPure XPsystem (Beckman Coulter, Beverly, MA, USA). Subsequently, library quality was assessed using the Agilent 5400 system (AATI) and quantified by real-time PCR (1.5 nM). The qualified libraries were pooled and sequenced on Illumina platforms with PE150 strategy in Novogene Bioinformatics Technology Co., Ltd. (Beijing, China), according to the effective library concentration and the data amount required.

### 2.5. Metagenomic Sequence Analysis

Intestinal content samples from the ileum at 14, 21, and 28 days of age were selected for metagenomic sequencing. Two low-quality samples were excluded, leaving sixteen samples for subsequent analyses. Raw sequencing data were quality-controlled using kneaddata (v0.12.2) [25] with the pig reference genome (Sus scrofa 11.1, GCF_000003025.6_Sscrofa11.1). Cleaned reads were assembled using MEGAHIT (v1.2.9) [26], and contigs were then processed with Prodigal (v2.6.3) [27] to predict functional genes. To remove redundancy, genes were dereplicated with CD-HIT (v4.8.1) [28] using the following parameters: -c 0.95 -G 0 -aS 0.9 -g 1. Gene quantification was performed with Salmon (v1.10.3) [29] after index construction.

For differential analysis, normalized read counts (numreads) from the three time points were analyzed in RStudio (version 4.4.2, 2024) using DESeq2 [30]. Differentially abundant genes were subsequently subjected to KEGG pathway enrichment analysis using the enricher function [31], and enriched functional modules were mapped via the KEGG Mapper tool (https://www.kegg.jp/kegg/mapper/reconstruct.html, accessed on 22 October 2025). Finally, differentially abundant genes were taxonomically assigned using Kraken2 (v2.1.3) [32] against the standard database to identify their host species.

## 3. Results

### 3.1. The 16s rRNA Gene Analysis Reveals the Important Role of Lactobacillus in Ileal Microbiota Dynamics of Piglets During the Nursing Period

We examined the species composition at different key time points in the ileum using 16s rRNA gene analysis. From 7 days post-birth, the gut microbiota of the ileum was primarily dominated by *Lactobacillus* and *Limosilactobacillus*, which together accounted for over 75% of the microbiota (Figure 1a).

To observe the changes in the gut microbiota of DLY piglets from birth to the appropriate weaning age, we conducted a statistical analysis of 16s rRNA gene data from the ileum. The Simpson index indicated no significant differences (*p* > 0.05) in the diversity of gut microbiota in the ileum across different growth stages (Figure 1b). Principal Coordinates Analysis (PCoA) of the ileum revealed similar trends, showing minimal beta diversity changes in the microbiota from 7 days after birth (Figure 1c). LEfSe was used to identify dominant bacterial taxa at each key time point. Comparisons between consecutive time points revealed that each 7-day interval after birth was associated with changes in one or more *Lactobacillus* (LDA score > 4.0).

In the ileum, compared to D1, *Lactobacillus johnsonii* (LDA = 5.37, *p* < 0.01) and *Lactobacillus delbrueckii* (LDA = 5.09, *p* < 0.05) were enriched at D7. At D7 vs. D14, *Ligiactobacillus salivarius* (LDA = 4.12, *p* < 0.01), *Limosilactobacillus pontis* (LDA = 4.11, *p* < 0.05), and *Limosilactobacillus coleohominis_A* (LDA = 4.04, *p* < 0.05) were enriched at D14. At D21 vs. D28, *Limosilactobacillus pontis* (LDA = 4.12, *p* < 0.01) was enriched at D21 (Figure 1d). In addition, *Akkermansia_muciniphila_D_776786* was enriched in D14 (D7 vs. 14) and D21 (D21 vs. 28).

These findings highlight the important role of *Lactobacillus* in the growth and development of piglets. Specifically, we observed significant changes in the relative abundance of *Lactobacillus* species at key time points. *Lactobacillus johnsonii* (Figure 1e) and *Lactobacillus delbrueckii* (Figure 1f) had higher relative abundances. At 14 days post-birth, *Lactobacillus delbrueckii* experienced a sharp decline. Meanwhile, *Ligiactobacillus salivarius* began to increase in relative abundance at 14 days (Figure 1g). At 21 days, *Limosilactobacillus pontis* had a high relative abundance (Figure 1h).

### 3.2. Metagenomic Analysis of Microbial Functional Changes in Piglets Aged 14–28 Days

The gut microbiota of piglets begins to stabilize around day 21 of lactation, a period critical for the functional maturation of the ileum. Therefore, we conducted metagenomic sequencing on samples at the ages of 14, 21, and 28 days. Differential abundance analysis (DESeq) revealed distinct functional profiles. At day 21, the top 10 enriched KEGG Orthologies (KOs) among upregulated genes were K03328, K00244, K01193, K15780, K02818, K02819, K03319, K05823, K20342, and K02817 (Figure 2a). In contrast, downregulated genes were predominantly enriched in K00016, K03293, K03778, K03294, K08659, K00841, K00712, K04086, K22393, and K01577 (Figure 2b). At day 28, the major enriched KOs for upregulated genes included K03293, K00016, K03294, K07052, K08659, K07493, K11733, K03758, K22393, and K18923 (Figure 2c), while those for downregulated genes were K01421, K00656, K21449, K00244, K03091, K06409, K18346, K04076, K07321, and K06399 (Figure 2d).

KEGG pathway enrichment analysis further showed that at day 21, upregulated genes were significantly enriched in the citrate cycle (TCA cycle), galactose metabolism, starch and sucrose metabolism, butanoate metabolism, C5-branched dibasic acid metabolism, oxidative phosphorylation, other carbon fixation pathways, purine metabolism, and peptidoglycan biosynthesis. Pathways enriched in downregulated genes included glycolysis/gluconeogenesis, the pentose phosphate pathway, pentose and glucuronate interconversions, glyoxylate and dicarboxylate metabolism, propanoate metabolism, cysteine and methionine metabolism, teichoic acid biosynthesis, glutathione metabolism, and riboflavin metabolism (Figure 2e).

At day 28, upregulated genes were significantly enriched in pathways such as glycolysis/gluconeogenesis, the pentose phosphate pathway, lysine biosynthesis, cysteine and methionine metabolism, and riboflavin metabolism. Conversely, downregulated genes were enriched in the citrate cycle (TCA cycle), butanoate metabolism, oxidative phosphorylation, and other carbon fixation pathways. Notably, a comparison across the three time points highlighted that the citrate cycle (TCA cycle), butanoate metabolism, oxidative phosphorylation, and other carbon fixation pathways were enriched at day 21 (showing upregulation compared to day 14 and downregulation compared to day 28) (Figure 2f).

Butanoate metabolism plays an important role in the gut health of piglets around 21 days of age. At day 21, genes enriched in butanoate metabolism are mainly carried by *Streptococcus suis*, *Streptococcus oriscaviae*, *Streptococcus parasuis*, *Streptococcus pluranimalium*, *Streptococcus equinus*, *Streptococcus gallolyticus*, *Lactococcus cremoris*, *Lactobacillus delbrueckii*, *Lactobacillus amylovorus*, *Limosilactobacillus pontis*, *Limosilactobacillus oris*, *Limosilactobacillus mucosae*, and *Carnobacterium maltaromaticum* (21 vs. 14 upregulated). At day 28, genes enriched in Butanoate metabolism are mainly carried by *Streptococcus suis*, *Streptococcus oriscaviae*, *Streptococcus* sp. 29887, *Streptococcus* sp. 29892, *Streptococcus parasuis*, *Streptococcus chenjunshii*, *Lactobacillus delbrueckii*, *Limosilactobacillus pontis*, *Ligilactobacillus salivarius*, and *Rothia nasimurium* (28 vs. 21 downregulated). The genes enriched in butanoate metabolism are mainly carried by *Streptococcus*, *Lactobacillus*, and *Limosilactobacillus* (Appendix A).

In D28, among the upregulated genes in the amino acid metabolism category, the major carriers of upregulated genes in the cysteine and methionine metabolism pathway were *Limosilactobacillus mucosae*, *Limosilactobacillus oris*, *Limosilactobacillus pontis*, *Lactobacillus helveticus*, *Lactobacillus crispatus*, and *Lentilactobacillus laojiaonis*. The major carriers of upregulated genes in the lysine biosynthesis pathway were *Limosilactobacillus reuteri*, *Limosilactobacillus pontis*, *Limosilactobacillus mucosae*, *Limosilactobacillus oris*, and *Streptococcus canis* (Appendix A).

## 4. Discussion

We profiled the ileal microbiota across early life and observed stable ileal α-diversity and limited β-diversity separation from day 7 to 28. From day 7, *Lactobacillus* and *Limosilactobacillus* constituted major components. Shotgun metagenomics at day 14, 21, and 28 indicated ileum functional patterns. Notably, the ileal metagenome at day 21 was significantly enriched in genes related to butanoate metabolism (e.g., K00244, K01575, and K00656). These results define a critical developmental window between days 14 and 28, characterized by coordinated taxonomic succession and functional maturation of the ileal microbiota.

Throughout the nursing transition, the succession of *Lactobacillus* reflected adaptation to nutritional, immune, and environmental changes. While *Lactobacillus* and *Limosilactobacillus* dominated from day 7 onwards (exceeding 75% relative abundance in the ileum), their dynamic changes were closely associated with microbial stabilization and host development. LEfSe analysis highlighted the dynamic role of *Lactobacillus* across developmental stages. Specifically, *Lactobacillus johnsonii* and *Lactobacillus delbrueckii* were enriched in the ileum at day 7. *Lactobacillus johnsonii* and *Lactobacillus delbrueckii* are typically used to reduce the occurrence of diarrhea in piglets and enhance intestinal barrier proteins [33,34]. *Ligiactobacillus salivarius* was enriched at day 14. *Ligiactobacillus salivarius* can increase the feed intake of piglets, thereby improving their growth performance [35]. *Limosilactobacillus pontis* was enriched at day 14 and 21. There have been relatively few interventions and functional tests for *Limosilactobacillus pontis* in animals. It may be related to some key functions of piglet development, and it is a potential probiotic.

During the nursing period from day 21 to day 28, ileum function became more complete. The shifts in upregulated and downregulated metabolic pathways in the ileum reflected the adaptability and functional diversity of the intestinal microbiota.

A key functional shift observed at day 21 was the significant enrichment of the butanoate metabolism pathway. Butyrate supplementation has been shown to improve the intestinal morphology by increasing villus height and crypt depth in the jejunum and ileum, thereby enhancing nutrient absorption and gut barrier function [36]. Butyrate is essential for providing energy to colonocytes, enhancing intestinal barrier function, and modulating host immune responses [37,38]. Notably, this functional enrichment corresponded with a taxonomic shift identified by LEfSe analysis, which revealed *Limosilactobacillus pontis* as a biomarker species significantly enriched at day 21. To mechanistically link this taxon with the observed function, we performed taxonomic annotation (Kraken2) on the enriched genes. This analysis confirmed that genes involved in butanoate metabolism were predominantly contributed by *Streptococcus*, *Lactobacillus*, and *Limosilactobacillus*, with *Limosilactobacillus pontis* being one of the key contributors. This finding is particularly relevant as day 21 is a critical developmental window often coinciding with early weaning. Butyrate supplementation has been associated with an increase in the abundance of beneficial gut bacteria such as Lactobacillus and Bifidobacterium, which play a role in maintaining gut health and preventing dysbiosis [39,40,41]. These findings indicate that the *L. pontis* plays a crucial role in enhancing butyrate synthesis, thereby supporting gut maturation and homeostasis during the critical pre-weaning transition as a potential probiotic.

At day 28, in addition to being enriched in carbohydrate metabolism, upregulated genes in the ileum were also significantly enriched in amino acid metabolism pathways, such as cysteine and methionine metabolism and lysine biosynthesis. The activation of these pathways may be associated with increased demands for energy and amino acids in the gut [42]. Lysine, cysteine, and methionine are restrictive essential amino acids during the growth process of piglets, which are important for optimal growth performance to meet the demands of rapid growth, immune regulation, and antioxidant defense [43,44,45]. Previous studies have indicated that early post-weaning piglets face multiple challenges, including incomplete intestinal structural maturation, rapid expansion of the immune system, and temporarily reduced nutrient absorption efficiency [46]. Our findings confirm this pattern: even before weaning, the gut microbiota of 28-day-old nursing piglets appeared to be more orientated toward supporting amino acid metabolism and tissue proliferation than at day 21, suggesting that the gut microbiota was functionally shifting toward “nutrient synthesis and growth support”. This can be viewed as a physiological adaptation in which the intestinal ecosystem and host metabolic demands evolve in synchrony. In practical applications, when the aim is to improve piglet health and promote weight gain through microbial regulation or targeted metabolic intervention, utilizing the *Lactobacillus* identified in this study, based on functional metagenomic analysis of healthy nursing piglets, may produce more effective results.

Microbial taxa played important roles in this functional differentiation. In the ileum, lactic acid bacteria such as *Limosilactobacillus* and *Lactobacillus* were involved in amino acid biosynthesis, involving lysine production and the metabolic processing of cysteine and methionine. The dynamic transition from day 14 to day 28 highlights how the ileal microbiota adjusts to changing metabolic demands during piglet growth and development.

## 5. Conclusions

The spatial–temporal dynamics of the gut microbiota in piglets were revealed through 16s rRNA gene sequencing. During the nursing period, there were minimal changes in species diversity and richness (α-diversity) in the ileum. Several species of *Lactobacillus* played key roles at different developmental time points, such as *Lactobacillus johnsonii*, *Lactobacillus delbrueckii*, *Ligilactobacillus salivarius*, and *Limosilactobacillus pontis*. Additionally, metagenomic analysis was used to investigate the enrichment of differential genes and metabolic pathways in the ileum between 14 and 28 days of age. *Limosilactobacillus pontis* affects the butanoate metabolism pathway in 21-day-old piglets as a potential probiotic.

## Figures and Tables

**Figure 1 animals-15-03102-f001:**
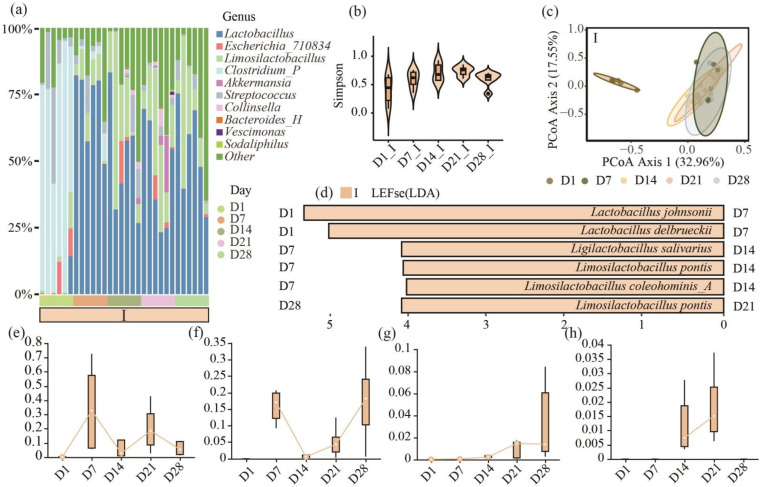
Analysis of key ages in the intestinal segment of piglets during lactation. (**a**) The composition of the ileum microbiota (at the phylum level); (**b**) α diversity analysis (Simpson index), with the light apricot color representing the ileum; (**c**) β diversity analysis (Bray–Curtis); (**d**) LEfSe analysis. Comparison of two adjacent key ages. The column is close to the enriched age. I, ileum (LDA > 4 indicates extremely significant difference). (**e**–**h**) Box plot of relative abundance changes in *Lactobacillus* with significant differences in key ages: *Lactobacillus johnsonii* (**e**); *Lactobacillus delbrueckii* (**f**); *Ligiactobacillus salivarius* (**g**); and *Limosilactobacillus pontis* (**h**).

**Figure 2 animals-15-03102-f002:**
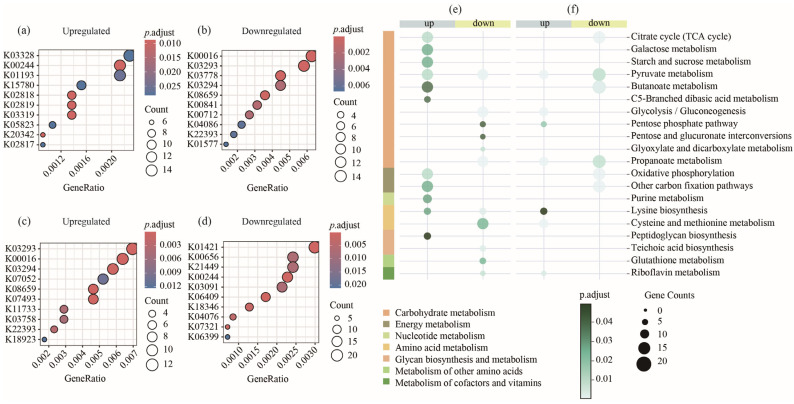
Analysis of differential gene enrichment in piglets. (**a**,**b**) KEGG functional units with up- and downregulated gene enrichment in the ileum at 21 days; (**c**,**d**) KEGG functional units with up- and downregulated gene enrichment in the ileum at 28 days; (**e**) metabolic function mapping of KEGG functional units with differential gene enrichment in the ileum at 21 days; (**f**) metabolic function mapping of KEGG functional units with differential gene enrichment in the ileum at 28 days.

## Data Availability

The dataset supporting the findings of this study is available in the NCBI repository, BioProject: PRJNA1330193.

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
