# Peer review of "Functional and Compositional Changes in Ileal Microbiota in Piglets During the Nursing Period Revealed by 16s rRNA Gene and Metagenomics"

_animals, 2025, doi:10.3390/ani15213102_

Round 1

Reviewer 1 Report

Comments and Suggestions for Authors

Dear authors,

thank you very much for submitting your manuscript. Please finde the reviewer's report down below.

Kind regards.

Reviewer's report

Summary:

In this study the authors examine the microbial composition and the differential gene expression in the ileum of suckling piglets on different days of life. They were able to show a different composition of Lactobacillus spp., Ligilactobacillus spp., and Limolactobacillus spp. at different times of examination. Differentially expressed genes indicate differences in butyrate, citrate, carbohydrate and other metabolic pathways.

General comment on the hypothesis of the work:

The present study is very interesting and provides important information on the composition of the microbiota in connection with metabolic pathways in the ileum during suckling phase.

Unfortunately, an ethical statement on the conduct of the animal experiment, including details of the authorizing authority, is missing. If this information cannot be provided, publication of the manuscript must be rejected.

The age of the piglets should be emphasized more clearly in the title of the paper. Better: ‘Functional and compositional changes of ileal microbiota in suckling piglets during the nursing period revealed by 16s rRNA gene and metagenomics’

However, in the materials and methods section, further additions must be made to the sows, piglets, husbandry and health status of the pigs. In addition, the performance of the animal experiment must be described in more detail.

In the results the formatting of the figures should be improved.

The discussion is short and precise. However, influences of husbandry and health status of the piglets in relation to the experiment conducted must be discussed with reference to data in the literature.

The citation style must be adapted throughout the manuscript. According to the journal’s guidelines, citations must be placed in square brackets at the end of a sentence. Please compare the citation style in the reference list with the guidelines of the journal. The literature sources were not listed in accordance with the specifications.

Comments on the Abstract:

L20: Lactobacillus spp.

Comments on Introduction:

L48: better: days of life

L50: days

Comments on Material and Methods:

L59: Material and Methods

LL61-62: Please add a description of husbandry of sows their offsprings, health status (especially clinical signs of PPDS in sows and diarrhea in piglets), treatments, vaccination and hygienic status of the pens. How many sows with their offspring were included in the experiment?

L63: Please add the total number of piglets included.

L64: euthanization instead of slaughter. Please describe the method of euthanization and pathological examination.

LL64-66: Which samples were collected? Mucosa of the ileum e.g. with a cell lifter? Please describe the method in more detail.

L66: Please add the manufacturer, city and state for the centrifuge tubes.

L79: Please add the manufacturer, city and state for QIIME2.

L89: Please add the manufacturer, city and state for the reference database.

LL92-93: Please add the city and federal state for Covaris LE220R-plus.

LL95-96: Please add the federal state for AMPure XPsystem

L96: Please add the manufacturer, city and state for the Agilent 5400 system.

L96: assessed on (missing space)

L105: Please add the manufacturer, city and state for kneaddata.

LL106-107: Please add the manufacturer, city and state for MEGAHIT.

L107: Please add the manufacturer, city and state for Prodigal.

L108: Please add the manufacturer, city and state for CD-HIT.

L110: Please add the manufacturer, city and state for Salmon.

L112: Please add the manufacturer, city and state for the statistical program R.

L116: Please add the manufacturer, city and state for Kraken2.

Comments on the Results:

L118: Results

L119: Lactobacillus spp.

L123: Lactobacillus spp.; Limosilactobacillus spp.

L125: Please explain the abbreviation DLY in L61.

L127: p < 0.05 (missing space)

L133: Lactobacillus spp.

L134: LDA = 5.37; p < 0.05 (missing space)

L135: LDA = 5.09; p < 0.05 (missing space)

L136: Ligilactobacillus

L136: LDA = 4.12; p < 0.05; LDA = 4.11; (missing space)

L137: LDA = 4.04; p < 0.05 (missing space)

L138: LDA = 4.12; p < 0.05 (missing space)

L141: Lactobacillus spp.

L143: Lactobacillus spp. instead of Lactobacillus species

L150: Simpson index

L152: key ages. The (missing dot)

L152: Ileum

L152; LDA > 4 (missing space)

L154: Lactobacillus spp. (italic style)

LL170-204: citrate, galactose; starch; butanoate; oxidative; other; purine; peptidoglycan; glycolysis; gluconeogenesis; pentose; glyoxylate; propanoate; cysteine; teichoic; glutathione, lysine; riboflavin

L201: Lactobacillus crispatus, and Lentilactobacillus

L206: gene expression instead of gene enrichment

Figure 1:

Please separate the figures and present in higher size.

In Fig. 1a the genus of different microbiota are listed. Please add the abbreviation spp. to the genus.

Please add headlines to the different figures.

Figure 2:

Please separate the figures and present in higher size.

Please add headlines to the different figures.

 Comments on the discussion:

L214: Lactobacillus spp.

L215: Limosilactobacillus spp.

L221: Lactobacillus spp.

L222: Lactobacillus spp.

L223: Limosilactobacillus spp.

L225: Lactobacillus spp.

L229: Ligilagtobacillus salivarius

L230: performance and (missing space)

L238: butanoate

L247: butanoate

LL247-248: Streptococcus spp., Lactobacillus spp., and Limosilactobacillus spp.

LL251-252: Lactobacillus spp.; Bifidobacterium spp. (italic style)

L253: L. pontis  (italic style) If you use the abbreviation for the genus, please explain it in previous lines.

L256: carbohydrate

LL264: Limosilactobacillus spp. and Lactobacillus spp.

L265: lysine

L272: Lactobacillus spp.

L276: butanoate

Comments on Supplemental Material:

The species in the table must be written in italic style.

Comments on the Quality of English Language

The manuscript should be edited for language. Some formulations are not clear and use of terms that are not commonly used. In addition, capitalisation needs to be improved.

Author Response

Thank you for your thoughtful suggestions on our study. We have revised the manuscript accordingly. The main changes are as follows:

  1. Section 2.1, Sample collection: We added an animal ethics statement and provided detailed descriptions of the samples.

  1. Scientific names: All species names in the main text and Supplementary Materials are now italicized. Due to the journal’s font, italics may appear subtle; metabolic pathway names that are common nouns are not italicized.

  1. Typography of symbols: We inserted spaces around operators such as “=”, “<”, and “>” throughout the manuscript, as recommended.

  1. Software citations and versions: For all software used (e.g., QIIME 2), we now provide the exact version numbers and corresponding citations, in accordance with the journal’s requirements.

  1. Figures: Figure 1 is presented at the genus level, which is our intended resolution and consistent with common practice; adding “pp.” is unnecessary. Each panel (e.g., a, b) is clearly labeled with corresponding information in the figure captions, and the main figure includes a title. Font sizes conform to the journal’s specifications.

  1. Abbreviations: The term DLY (Duroc × Landrace × Yorkshire) is now defined at its first occurrence in the Sample collection section.

  1. References: We have revised the formatting of all references to match the journal’s style.

  1. Sample-related discussion: We expanded the discussion of sample characteristics (Lines 294–305).

In this study, we focused on the temporal dynamic changes of the ileal microbiota and combined 16S rRNA gene sequencing with metagenomic sequencing to provide a unique perspective for the development of intestinal microbiota in piglets. In terms of sample selection, because our study aimed to investigate temporal variation, it was essential to keep other conditions consistent among piglets during the suckling period, such as breed, environment, management, and diet, allowing time to be the only variable influencing the ileal microbiota. Therefore, samples were collected from piglets of the same breed at different ages for comparative analysis. Thus, potential probiotics under the normal growth and development conditions of nursing piglets can be screened out. Limitation in this study is focused solely on the temporal variation within the ileum and did not address spatial differences among multiple intestinal segments, due to practical constraints.

Reviewer 2 Report

Comments and Suggestions for Authors

1) The result of this analysis may be common sense, and there are many similar results in literature reported, including those references quoted in the text. 
2) The author only chose the ileam section, and the piglets in trial seemed no difference from regular feeding, including breed, surrounding, management and diet.
3)  Focus on segment-resolved and time-resolved, or the spatial-temporal dynamics of the gut microbiota in piglets, which could not be reflected by the expriment at all, except different  sampling time, and the piglets and ileam sections have no special.
4) As for the 16s rRNA gene sequence analysis and  the metagenomic sequence analysis, they are       the frequently-used methods in bio-informatics, particularly popular in anmal intestinal microbita research.
5) As well-known, Lactobacillus played main roles in gut. Many metabolic pathways have been identified so far, including benifical butanoate metabolism.

Author Response

Thank you for your thoughtful suggestions on our study:

Indeed, numerous studies have investigated the gut microbiota of piglets, most of which focus on fecal samples to assess the intestinal health of nursing piglets, or analyze a specific intestinal segment at a single time point. However, each intestinal segment plays distinct physiological roles within the gut. This is particularly evident during the nursing period, when both the proximal and distal intestines undergo continuous development and maturation, each contributing differently to digestion and nutrient absorption. In this study, we focused on the temporal dynamics of the ileal microbiota and combined 16S rRNA gene sequencing with metagenomic sequencing, providing a unique perspective on the gut microbial development of piglets.

Similarly, in terms of sample selection, because our study aimed to investigate temporal variation, it was essential to keep other conditions consistent among piglets during the suckling period, such as breed, environment, management, and diet, allowing time to be the only variable influencing the ileal microbiota. Therefore, samples were collected from piglets of the same breed at different ages for comparative analysis. Thus, potential probiotics under the normal growth and development conditions of nursing piglets can be screened out. However, due to practical constraints, this study focused solely on the temporal variation within the ileum and did not address spatial differences among multiple intestinal segments—an acknowledged limitation that we aim to address in future research.

Although 16S rRNA gene sequencing and metagenomic sequencing are commonly used methods in animal gut microbiome studies, metagenomic sequencing requires a much higher amount of DNA input. During the nursing period, piglets rely exclusively on maternal milk, making intestinal content sampling technically challenging. The small intestine, in particular, contains a very limited amount of material, resulting in most samples failing to meet the quality requirements for metagenomic sequencing, though they remain suitable for 16S rRNA gene sequencing. To overcome this limitation, our study first employed 16S rRNA gene sequencing to identify the critical time points of microbial changes in the ileum, followed by metagenomic analysis for in-depth functional exploration. This approach effectively compensates for the scarcity of intestinal content and ensures robust biological interpretation.

In the functional analyses, many Lactobacillus have previously been shown to participate in metabolic pathways such as butyrate metabolism. However, our integrated analysis revealed significant differences in the relative abundance of different Lactobacillus at various ages, suggesting that these species may perform distinct functional roles. Subsequent metagenomic functional analyses further confirmed that Limosilactobacillus pontis plays an important role in specific metabolic pathways, supporting this hypothesis. Since metagenomic analysis enables species-level resolution, we identified L. pontis as a potential key probiotic species contributing to ileal development in nursing piglets. Although research on L. pontis remains limited, our findings highlight its potential importance in promoting gut maturation, which underscores the significance of this study.

Reviewer 3 Report

Comments and Suggestions for Authors

The study describe the changes of microbiota in piglets between 1 to 28 days using 16s rRNA gene sequencing methods to identify changes in the microbiome. At days 21 and 28, major changes where identify associated with Limosilactobacillus mucosae, Limosilactobacillus oris, and Limosilactobacillus pontis.

There is some questions about how the experiment was performed, specially in the number of piglets, if the same piglets were used all the days and so on. Please provide more information.

Author Response

  1. Thank you for your questions and suggestions. Our previous description of sample collection was not sufficiently clear. To clarify, we slaughtered piglets of the Duroc × Landrace × Yorkshire (DLY) cross at different time points and collected ileal samples to analyze the temporal dynamics of the ileal microbiota. We have also added the details of approval by the institutional animal ethics committee. These additions have been incorporated into the Sample collection section (Lines 73–88):

This study used a crossbred piglet model DLY (Duroc × Landrace × Yorkshire) under uniform feeding conditions in Guangzhou, China (The entire breeding period is managed in accordance with the regular immunization and health care procedures of the pig farm). Piglets were slaughtered at five time points and intestinal contents samples were collected then used for analyzing the temporal dynamics of the ileum: 1, 7, 14, 21, and 28 days of age, with six piglets per time point (A total of 30 piglets). To ensure consistency, piglets with similar body weights from different stalls were selected for slaughter, maintaining a 1:1 male-to-female ratio (To reduce the need for additional repetitive experiments due to extreme weight or gender differences, and to minimize stress and pain to the greatest extent possible). Upon dissection, the ileum were carefully separated, and intestinal contents were collected from the mid-segment of ileum using 15 mL centrifuge tubes. Immediately following collection, intestinal contents were placed on liquid nitrogen for rapid freezing. All animal work was conducted according to the guidelines for the care and use of experimental animals established by the Ministry of Agriculture of China. The project was also approved by Experimental Animal Ethics Committee of Foshan University (Number: FOSU2019029906).

  1. Thank you for your suggestion. We have supplemented the metagenomic approach section (Line 122-125) :

Based on the results of 16s RNA gene analysis, we found that the intestinal microbiota was stable at 21 days of age. We believe that this period plays an important role in the functional maturation of the ileum. Therefore, a total of 18 samples from the ileum at 14, 21 and 28 days of age were subjected to metagenomic sequencing.

  1. The discussion on Lactobacillus at different time points in the ileum is explained on Line 244-257

  1. The presentation form of Figure1c complies with the requirements of β diversity analysis (Bray-Curtis), which is used to observe the degree of dispersion between each group.

Round 2

Reviewer 1 Report

Comments and Suggestions for Authors

Dear authors,

thank you very much for incorporating the most of my comments. I have only a few further comments. Please finde my comments down below.

Kind regards.

Comments:

L19: analysis

L45: dys [3] (missing space)

L49: butyrate [7] (missing space)

L51: immunity [8] (missing space)

L54: life [11] (missing space)

L55: assembly [15] (missing space)

L64: profiling [18] (missing space)

LL71, 98, 121, 179: Please use uniform capitalization in the headlines.

LL73-74: Which vaccinations in detail were used, especially intestinal agents?

L99: (version 2023.9.1) [22] (missing space)

L109: database [23] (missing space)

L127: (v0.12.2) [24] (missing space)

L129: (v1.2.9) [25] (missing space)

L130: (v2.6.3) [26] (missing space)

L131: (v4.8.1) [27] (missing space)

L132: (v1.10.3) [28] (missing space)

L134: DESeq2 [29] (missing space)

L136: function [30] (missing space)

L138: (v2.1.3) [31] (missing space)

L253: performance and [34] (missing space)

L262: function [35] (missing space)

L264: responses [36] (missing space)

L274: dysbiosis [38] (missing space)

L282: gut [41] (missing space)

L284: defense [42] (missing space)

Please check my earlier comments of the first review on Material and Methods. These comments aren´t included yet:

LL64-66: Which samples were collected? Mucosa of the ileum e.g. with a cell lifter? Please describe the method in more detail.

L66: Please add the manufacturer, city and state for the centrifuge tubes.

L79: Please add the manufacturer, city and state for QIIME2.

L89: Please add the manufacturer, city and state for the reference database.

LL92-93: Please add the city and federal state for Covaris LE220R-plus.

LL95-96: Please add the federal state for AMPure XPsystem

L96: Please add the manufacturer, city and state for the Agilent 5400 system.

L96: assessed on (missing space)

L105: Please add the manufacturer, city and state for kneaddata.

LL106-107: Please add the manufacturer, city and state for MEGAHIT.

L107: Please add the manufacturer, city and state for Prodigal.

L108: Please add the manufacturer, city and state for CD-HIT.

L110: Please add the manufacturer, city and state for Salmon.

L112: Please add the manufacturer, city and state for the statistical program R.

L116: Please add the manufacturer, city and state for Kraken2.

Author Response

Thank you for your thoughtful suggestions on our study. We have revised the manuscript accordingly. The main changes are as follows:

1.We have supplemented the spaces in all references and modified the reference style.

  1. Headings have been uniformly capitalized.

3.Our descriptions may have caused misunderstandings and have been revised accordingly.

L73-74: (The entire breeding period is managed in accordance with the regular procedures of the pig farm).

  1. Regarding sample collection queries, in intestinal research, the term "intestinal content" simply refers to mixtures obtained directly from within the intestines.It does not refer to the "Mucosa of the ileum." We have provided a more detailed description of the sampling procedure:

L81-85: Upon dissection, the ileum were carefully separated. After separating the middle section of the ileum, gently squeeze the intestinal content into the centrifuge tubes (Servier (Beijing) Pharmaceutical R&D Co., Ltd.) to avoid contact with oxygen. Immediately following collection, intestinal content were placed on liquid nitrogen for rapid freezing.

  1. Tools such as QIIME 2, kneaddata, and Prodigal are considered "methodological literature" that require citation, rather than "commercial products" that require manufacturer identification. In accordance with journal requirements, the names and versions of the tools have been clearly stated in the Methods section.

Reviewer 3 Report

Comments and Suggestions for Authors

The authors answered partially the comments of the past review, which reflect in some errors of this review.

My major concern is about the handling of the animals, first you have to provide information of the ethical approval committee, as it is stablished in the Ethical Guideline for the use of animals research in the instructions for authors.

You must address the issue about the reduction of the sample number, it's distribution among the time periods and the female/male ratio in the final samples (16 selected). No addressing these issues is a violation of the ARRIVE guidelines 2.0 which MDPI endorse for publications with animal experimentation.

Author Response

Thank you for your thoughtful suggestions on our study. We have revised the manuscript accordingly. The main changes are as follows:

  1. L74-78 : Under the guidance of professional veterinarians, piglets were slaughtered at five time points and intestinal contentsamples were collected then used for analyzing the temporal dynamics of the ileum: 1, 7, 14, 21, and 28 days of age, with six piglets per time point (A total of 30 piglets).
  2. The Institutional Review Board Statement has been supplemented and uploaded

L312-315:All animal work was conducted according to the guidelines for the care and use of experimental animals established by the Ministry of Agriculture of China. The project was also approved by Experimental Animal Ethics Committee of Foshan University (Number: FOSU2019029906).  

  1. The number of piglets in this study has not changed. The description of the Metagenomic Sequence Analysis method and results may cause confusion. The samples used in the metagenomic analysis refer to the same intestinal contentfrom the ileum as those used in the 16sRNA gene analysis, rather than different piglets. This is as stated in the introduction of this study: A two-tier design incorporated shotgun metagenomics at day 14-21 and 21-28 to compare gene repertoires and KEGG-annotated pathways. The approach accommodated low microbial biomass in early life and the higher DNA input typically required for shotgun metagenomics relative to 16s rRNA gene profiling. The 16sRNA gene analysis and Metagenomic Sequence Analysis both used the intestinal content samples from the same group of piglets. In this study, based on the results of 16sRNA gene analysis, we explored the development patterns of the ileal microorganisms over five time periods and concluded that the ileal microorganisms play an important role during the 21-day-old period. Therefore, through metagenomic sequencing, we conducted in-depth functional analysis of the ileal microorganisms of piglets at 14, 21, and 28 days of age. Therefore, this part of the method should be emphasized in the results.

The article has been modified as follows:

L123-125:Intestinal content samples from the ileum at 14, 21 and 28 days of age were selected to metagenomic sequencing. Two low-quality samples were excluded, leaving 16 samples for subsequent analyses.

L179-181: The gut microbiota of piglets begins to stabilize around day 21 of lactation, a period critical for the functional maturation of the ileum. Therefore, we conducted metagenomic sequencing on samples at the ages of 14, 21, and 28 days.

  1. For the method corresponding to Result 3.1, it has been supplemented in Section 2.3.16s rRNA Gene Sequence Analysis, Regarding the influence of the gender and days factors, this is indeed a limitation of this study. In future research, we will collect more samples to conduct related studies.

L118-121:Statistical tests for alpha (Simpson) and beta (Bray-Curtis) diversity were conducted using RStudio (version 4.4.2, 2024). The identification of characteristic microorganisms was achieved through LEfSe (Linear Discriminant Analysis Effect Size) analysis.

  1. For Figure1c, in the Pcoa diagram, the X-axis and Y-axis are the coordinates. This is a simple and clear Pcoa diagram